

# Prescriptive process monitoring: *Quo vadis?*

Kateryna Kubrak[1], Fredrik Milani[1], Alexander Nolte[1,2] and Marlon Dumas[1]

[1] Institute of Computer Science, University of Tartu, Tartu, Estonia
[2] Carnegie Mellon University, Pittsburgh, PA, United States of America

## ABSTRACT

Prescriptive process monitoring methods seek to optimize a business process by recommending interventions at runtime to prevent negative outcomes or address poorly performing cases. In recent years, various prescriptive process monitoring methods have been proposed. This article studies existing methods in this field via a systematic literature review (SLR). In order to structure the field, this article proposes a framework for characterizing prescriptive process monitoring methods according to their performance objective, performance metrics, intervention types, modeling techniques, data inputs, and intervention policies. The SLR provides insights into challenges and areas for future research that could enhance the usefulness and applicability of prescriptive process monitoring methods. This article highlights the need to validate existing and new methods in real-world settings, extend the types of interventions beyond those related to the temporal and cost perspectives, and design policies that take into account causality and second-order effects.

## INTRODUCTION

Process mining is a family of techniques that facilitate the discovery and analysis of business processes based on execution data. Process mining techniques use event logs extracted from enterprise information systems to, for instance, discover process models (*Agostinelli et al., 2019*) or to check the conformance of a process with respect to a reference model (*Van Der Aalst, 2012*). In this setting, an event log is a dataset capturing the step-by-step execution of a business process and includes timestamps, activity labels, case identifiers, resources, and other contextual attributes related to each case or each step within a case.

Over time, the scope of process mining has extended to encompass other use cases (*Milani et al., 2022*) such as techniques for predicting the outcome of ongoing cases of a process based on machine learning models constructed from event logs (*Maggi et al., 2014*; *di Francescomarino et al., 2018*). Predictions, however, only become useful to users when they are combined with recommendations (*Márquez-Chamorro, Resinas & Ruiz-Cortéz, 2018*). In this setting, prescriptive process monitoring is a family of methods that recommends interventions during the execution of a case that, if followed, optimize the process with respect to an objective (*Shoush & Dumas, 2021*). For instance, an intervention might improve the probability of the desired outcome (*e.g.*, on-time delivery) or mitigate negative

Corresponding author
Kateryna Kubrak,
kateryna.kubrak@ut.ee

outcomes (*e.g.*, delivery delays) (*Metzger, Kley & Palm, 2020*). Different implementations of prescriptive process monitoring have been proposed in the literature. However, the understanding of prescriptiveness in the field varies. To this end, some methods are guiding the user during the execution of the case based on its similarity to previous executions (*Terragni & Hassani, 2019*). Other methods specifically aim at optimizing process performance through correlation-based (*Gröger, Schwarz & Mitschang, 2014*; *Ghattas, Soffer & Peleg, 2014*) or causality-based predictions (*Bozorgi et al., 2021*; *Shoush & Dumas, 2021*). The methods also differ in other aspects, such as interventions prescribed. In some cases, two different methods aim at achieving the same objective but in different ways. For instance, to avoid an undesired outcome, one method might prescribe assigning resources for the next task (*Sindhgatta, Ghose & Dam, 2016*), whereas another might recommend which task to execute next (*de Leoni, Dees & Reulink, 2020*).

The benefits of prescriptive process monitoring can only be fully realized if these methods prescribe effective interventions that are followed (*Dees et al., 2019*). At present, though, the variety and fragmentation of prescriptive monitoring methods makes it difficult to understand which method is likely to be most effective or more likely to be accepted by end-users in a given business situation. There is no overview that captures existing prescriptive monitoring methods, what objectives they pursue, which interventions they prescribe, which data they require, or the extent to which these methods have been validated in real-life settings. Research overviews and classification frameworks have been put forward in the related field of predictive monitoring (*di Francescomarino et al., 2018*; *Márquez-Chamorro, Resinas & Ruiz-Cortéz, 2018*) and automated resource allocation (*Arias et al., 2018*; *Pufahl et al., 2021*). However, such works do not provide a structured overview of existing prescriptive process monitoring methods, nor serve researchers with a base for uncovering underserved but potentially valid areas of research in this field.

To address this gap, we study five research questions:

- Given that prescriptive process monitoring methods aim at prescribing interventions that produce business value, *i.e.,* achieve an objective, we formulate the first research question as: **RQ₁**. What is the objective for using prescriptive process monitoring methods to optimize a process?
- The second research question aims at discovering how these objectives can be achieved: **RQ₂**. What are interventions that are prescribed by prescriptive process monitoring methods?
- Third, we explore the data required by the proposed methods: **RQ₃**. What data do prescriptive process monitoring methods require?
- The fourth research question explores what modeling techniques the methods utilize to make use of the input data: **RQ₄**. What modeling techniques do prescriptive process monitoring methods use?
- Finally, we explore under which policy interventions are prescribed: **RQ₅**. What policies do prescriptive process monitoring methods use?

To answer these questions, we conducted a systematic literature review (SLR) following the guidelines proposed by *Kitchenham & Charters (2007)*. We identified 37 papers that we

analyzed to develop a multi-dimensional framework to characterize prescriptive monitoring methods. The contribution of this article is threefold. First, we provide a review of existing prescriptive process monitoring methods. Second, we develop a framework that classifies prescriptive process monitoring methods according to their objective, metric, intervention types, techniques, data inputs, and policies to trigger interventions. Third, we outline existing research gaps and provide insights into potential areas for future research in the field of prescriptive process monitoring. Our contribution aims at supporting researchers of prescriptive process monitoring methods. Researchers benefit from this contribution as they can gain insight into the current state of the art of the field and identify potential directions for future research. Developers of process mining tools who are interested in incorporating prescriptive process monitoring into their tools can also benefit from this work by better understanding the limitations and perspectives of existing methods.

The remainder of this article is structured as follows. First, we discuss 'Background and Related Work'. Then, we elaborate on the 'Method' and describe the 'Results'. We then propose a 'Framework' for prescriptive process monitoring methods, and conclude the paper in 'Conclusion'.

## BACKGROUND AND RELATED WORK

Process mining takes event logs of a system that supports the execution of a business process to discover process models (*Van Der Aalst, 2012*). An event log contains information, such as timestamps, activity, unique case id, resources, and other contextual attributes about executed cases. Thus, process mining finds its applications in the discovery of data-driven process models which can then be analyzed and improved. Process mining is also used for conformance checking which helps to evaluate whether a real process corresponds to a process model and process enhancement through which a process model can be enriched with additional information (*Van Der Aalst, 2012*), such as performance data (*Milani & Maggi, 2018*). With the advancement of technology, process mining has developed beyond process discovery, conformance, and enhancement (*Milani et al., 2022*). A large number of predictive process monitoring techniques have been proposed that are able to predict time-, risk-, and cost-related outcomes, as well as sequences of outcomes and inter-case metrics (*di Francescomarino et al., 2018*). These predictions can also further be used for prescriptive process monitoring.

Methods for prescriptive process monitoring prescribe interventions that can change the outcomes of an ongoing process case. For instance, if a method detects that an undesired outcome is probable to unfold, an alarm is raised that can lead to an intervention (*Teinemaa et al., 2018*). This intervention could *e.g.*, come in the form of an action performed by a process worker, such as calling a customer, that helps to mitigate or prevent the negative outcome from materializing (*Fahrenkrog-Petersen et al., 2022*). Thus, it is essential to define a policy for when a prescription is generated. The aforementioned example (*Fahrenkrog-Petersen et al., 2022*) *e.g.*, considers the probability of a negative outcome and evaluates the cost model and mitigation effectiveness before triggering interventions. In other words, to decide whether an intervention should be prescribed, this particular technique first checks

how probable it is that the ongoing case will end in a negative outcome. If the probability is high, it also calculates the trade-off between the cost and effect of intervening. Based on the outcome of the cost-effect analysis, this technique triggers an intervention.

Few prior studies focus on areas that are related to prescriptive process monitoring. For example, *di Francescomarino et al. (2018)* introduce a value-driven framework that allows companies to identify when to apply predictive process monitoring methods. The authors provide examples of predictive process monitoring techniques from the perspective of inputs that the techniques use, tools that implement them, and the domains where they had been tested. Another classification of predictive process monitoring methods has been proposed by *Márquez-Chamorro, Resinas & Ruiz-Cortéz (2018)*, who focus on methods to train predictive models. In particular, the authors describe building and evaluating a predictive model. In *Mertens et al. (2019)*, the authors evaluate predictive methods used to recommend follow-up activities in the healthcare domain. Thus, the focus there lies in describing the application of predictive process monitoring for processes prevalent in hospitals. While prescriptive process monitoring methods often incorporate predicted outputs, our work solely focuses on prescriptive methods and prescribed interventions.

In *Lepenioti et al. (2020)*, the authors review methods for prescriptive analytics. The authors provide a taxonomy of existing prescriptive analytics methods and map out research challenges and opportunities. *Pufahl et al. (2021)* present a systematic literature review on automatic resource allocation. The authors provide an overview of approaches with regards to resource allocation goals and capabilities, use of models and data, and their maturity. Similarly, *Arias et al. (2018)* give an overview of resource allocation methods, but with a particular focus on human resources. However, *Lepenioti et al. (2020)* reviews prescriptive methods in general, and the latter two (*Pufahl et al., 2021*; *Arias et al., 2018*) focus on resource allocation. In this article, we build on such works by considering various types of potential interventions in process-aware methods.

## METHOD

We conducted a systematic review of the existing body of work on prescriptive process monitoring methods. More specifically, what the objectives for using such methods are ($RQ_1$), what interventions such methods prescribe ($RQ_2$), what data such methods require ($RQ_3$), which modeling techniques they employed ($RQ_4$) and which policies ($RQ_5$) the methods use. The main objective of the SLR was, thus, to explore different aspects of prescriptive process monitoring. Therefore, we searched for methods and case studies that explore or study how ongoing process cases can be improved through the intervention of caseworkers. We use the method of SLR as it is particularly suitable for identifying relevant literature on a specific research topic (*Kitchenham & Charters, 2007*). We followed the guidelines proposed by *Kitchenham & Charters (2007)*, which consist of three main steps: (1) planning the review, (2) conducting it, and (3) reporting the findings.

For the first step (planning), we identified research questions and developed the review protocol (*Kitchenham & Charters, 2007*). Then, we developed a search string for the review protocol, identified suitable electronic databases, and defined inclusion and exclusion criteria. Finally, we defined the data extraction strategy.

In the search string, we included "process mining" to scope the study to methods that rely on event logs. We derived the term "prescriptive" from the research questions. We also included the terms "recommender" (*e.g.*, *de Leoni, Dees & Reulink, 2020*; *Yang et al., 2017*) and "decision support" (*e.g.*, *Mertens, 2020*), as we found these to be sometimes used instead of "prescriptive". Accordingly, we formulated the following search string:

*(recommender OR "decision support" OR prescriptive) AND "process mining"*

While conducting the first search, we noted that the term "prescriptive process monitoring" was not consistently used. Therefore, only using this search string might have resulted in missing relevant studies. We addressed this issue by examining the papers we identified using the first search string to identify other terms. We noted that terms such as "next-step recommendation" (*Huber, Fietta & Hof, 2015*), "next best actions" (*Weinzierl et al., 2020a*), "proactive process adaptation" (*Metzger, Kley & Palm, 2020*) have been used synonymously for "prescriptive process monitoring". We also noticed that the phrase "business process" often appeared in titles and keywords. Therefore, we formulated the following second search string:

*(recommender OR "next activity" OR "next step" OR "next resource" OR proactive) AND "business process"*

We applied both search strings to the ACM Digital Library, Scopus (includes SpringerLink), the Web of Science, and IEEE Xplore to identify potentially relevant papers. The databases were selected based on the coverage of publications within the field of process mining. Finally, we conducted backward referencing (snowballing) (*Okoli & Schabram, 2010*) to identify additional relevant papers.

Next, we defined exclusion and inclusion criteria (see Table 1). For the set of exclusion criteria, we excluded papers for which the answer to any of the defined exclusion criteria was "no": not digitally accessible (EC1), not in English (EC2), duplicates (EC3), and shorter than six pages (EC4). Exclusion criteria EC1 and EC2 ensured that the paper could be generally accessed and understood by other researchers. Papers that were unavailable *via* open access or *via* subscription services of the University, or *via* internet search were excluded. Papers in any other language than English were also removed. Criterion EC3 removed duplicates due to the paper being included in several digital libraries. We applied criterion EC4, as papers with less than six pages were not likely to contain sufficiently detailed information about the prescriptive process monitoring method for our analysis. We also defined three inclusion criteria: (IC1) the paper is relevant to the domain of prescriptive process monitoring, (IC2) the paper presents, reviews, discusses, or demonstrates a method or a case for prescriptive process monitoring, (IC3) the paper describes at least one way to identify candidate interventions for an ongoing process case. Thus, the answer for the inclusion criteria had to be "yes". IC1 aimed to include papers that are inside of the prescriptive process monitoring domain. With IC2, we made sure to include studies that represent a theoretical discussion or practical application of a method. Inclusion criterion IC3 ensured that the papers contained sufficient information to address our research questions. Following a top-down approach, if a paper failed an inclusion criterion, it was excluded without the other criteria being considered.

[1] The first search was conducted on 22 Sep 2021, the second on 12 Oct 2021.

[2] Full review protocol: https://doi.org/10.6084/m9.figshare.19455572.v2.

**Table 1  Exclusion/inclusion criteria utilized.**

| Criterion | Description |
| --- | --- |
| EC1 | The paper is digitally accessible. |
| EC2 | The paper language is English. |
| EC3 | The paper is not a duplicate. |
| EC4 | The paper is longer than six pages. |
| IC1 | The paper is relevant to the domain of prescriptive process monitoring. |
| IC2 | The paper presents, reviews, discusses, or demonstrates a method or a case for prescriptive process monitoring. |
| IC3 | The paper describes at least one way to identify candidate interventions for an ongoing process case. |

Finally, we defined our data extraction strategy (see Table 2). We first captured the metadata of all papers (title, author, publication venue, year). Then we defined the data required to address the research questions. Thus, for **RQ₁**, we defined the data required to identify the objective of using the prescriptive process monitoring technique and performance metric(s) targeted in each paper. Next, we defined the data to elicit the interventions prescribed, the process perspective, and the users the prescribed interventions are presented to (**RQ₂**). Then, we defined the required data input for the techniques described in the different papers (**RQ₃**). Finally, we added modeling techniques (**RQ₄**) and policies used to trigger interventions (**RQ₅**) to the data extraction strategy.

We executed the search [1] and identified a total of 1,367 papers (see Table 3). We filtered them using exclusion criteria EC1 and EC2. This resulted in the removal of 97 papers. Thus, 1,270 papers remained and were filtered based on EC3. Out of the remaining 1,010 papers, we removed short papers (EC4). This resulted in 900 papers remaining. These were filtered by title, thus removing papers that were clearly out of scope. The remaining 171 papers were filtered by abstract, resulting in 66 papers remaining. Finally, we applied the inclusion criteria by reading the whole paper and removed 44 papers. As a result, 22 papers remained. A total of 15 papers were added through backward referencing, resulting in the final list of 37 relevant papers.[2]

To derive the framework, we started by clustering the prescriptive process monitoring methods described in the identified papers according to what they were aiming to improve (**RQ₁**), *e.g.*, "cycle time minimization", "cost optimization". For each group, we followed the research questions to classify the methods further, such as according to the interventions they trigger (**RQ₂**), the input data they require (**RQ₃**), the modeling techniques they use (**RQ₄**), and the policies they utilize to trigger interventions (**RQ₅**).

## RESULTS

In the following subsections, we present the results of our review. First, we provide a quantitative overview of the identified papers. Next, we describe the identified objectives of prescriptive process monitoring methods (**RQ₁**) followed by the interventions prescribed to achieve these objectives (**RQ₂**). Next, we outline the data (**RQ₃**) and modeling techniques

**Table 2  Data extraction form.**

| Extracted data | Description |
|---|---|
| | Identification data |
| ID | Unique identifier of the paper |
| Title | Title of the paper |
| Author(s) | Authors of the paper |
| Year | Year of publication of the paper |
| Publication venue | Venue where the paper was published |
| | Study context |
| Process | Type of the process used in the example |
| Industry | The domain the dataset represents |
| Company | The company type in the domain the dataset represents |
| Dataset | Whether the dataset is real (taken from a real company) or synthetic (generated artificially) |
| | Prescriptive parameters |
| Intervention | Specific intervention prescribed |
| Process aspect | The process aspect (*e.g.*, control flow) for which the intervention is prescribed |
| Objective | Why the intervention is prescribed |
| Performance metric | Performance metric to measure the effectiveness of the prescribed intervention |
| For Whom | Who the intervention is prescribed for (e.g., process worker) |
| | Data & Technique |
| Input | Input data used in the method |
| Technique | Modeling technique used in the method |
| Policy | Policy used to prescribe the intervention |

**Table 3  Paper selection process.**

| Search | First | | Second | | Aggregated | |
|---|---|---|---|---|---|---|
| Selection criteria | # found | # left | # found | # left | # found | # left |
| Search results | 572 | | 795 | | 1,367 | |
| Data cleaning | 60 | 512 | 37 | 758 | 97 | 1,270 |
| Filtering by duplicates | 116 | 396 | 144 | 614 | 260 | 1,010 |
| Filtering by # of pages | 31 | 365 | 79 | 535 | 110 | 900 |
| Filtering by paper title | 252 | 113 | 477 | 58 | 729 | 171 |
| Filtering by paper abstract | 64 | 49 | 41 | 17 | 105 | 66 |
| Filtering by full paper | 34 | 15 | 10 | 7 | 44 | 22 |
| Backward referencing | 12 | | 3 | | | |
| **Total** | | 27 | | 10 | | **37** |

(**RQ₄**) used. Finally, we describe the different policies used by the prescriptive methods to trigger interventions (**RQ₅**).

## Quantitative overview

The distribution of the papers over years of publication is depicted in Fig. 1. Among the identified papers, we note the first one was published in 2008. Since then, studies on the topic were published each year. However, 20 out of 37 papers were published in the past five years. Thus, the years 2017–2019 produced three or four papers each, and in the year 2020, six papers were published. All identified papers were peer-reviewed, with eight of them being journal articles and 29 conference papers.

We note that the majority of papers use real-life event logs to validate their methods. As such, 19 papers utilize real-life event logs (among which are personally acquired logs and logs from the Business Process Intelligence Challenge (BPIC)), (https://www.tf-pm.org/competitions-awards/bpi-challenge) nine papers use synthetic logs, and three papers conduct validation on both real-life and synthetic logs. The choice of event logs also dictates the domain. For example, among the BPIC event logs, the most popular is the log from a financial institution (used in nine papers). Logs acquired personally are from manufacturing firms, public, and IT services. Synthetic event logs are generated for such domains as consultancy, electronics, manufacturing, and healthcare.

## Prescriptive process monitoring objectives

In our review, we identified two main objectives that prescriptive process monitoring methods aim to achieve. The first objective is related to optimizing the process outcome whereas the second concerns optimizing the process efficiency. We adopt this classification based on the definition of quality and efficiency of service by *Dumas et al. (2018)*. The objective of optimizing the process outcome relates to ensuring that the process outcome is positive. This objective is commonly expressed with binary metrics, such as avoiding a deadline violation (*Gröger, Schwarz & Mitschang, 2014*). The second objective of optimizing process efficiency relates to a particular quantitative aspect of process performance. Therefore, the objective of optimizing process performance can be expressed as, for instance, reducing cycle time (*Wibisono et al., 2015*).

As to the objective of *optimizing the process outcome*, five papers discuss undesired temporal outcomes, such as violation of a planned cycle time or deadline (*Gröger, Schwarz & Mitschang, 2014*; *Sindhgatta, Ghose & Dam, 2016*; *Weinzierl et al., 2020a*; *Huber, Fietta & Hof, 2015*; *de Leoni, Dees & Reulink, 2020*). For instance, *Gröger, Schwarz & Mitschang (2014)* describe an example from a manufacturing process, where the target is to reduce deadline violations. Thus, the possible outcomes can be described as binary: a deadline is violated (i) or a deadline is not violated (ii). Another set of studies focuses on avoiding or mitigating an undesired categorical outcome (*Teinemaa et al., 2018*; *Fahrenkrog-Petersen et al., 2022*; *Metzger, Kley & Palm, 2020*; *Shoush & Dumas, 2021*; *Ghattas, Soffer & Peleg, 2014*; *Thomas, Kumar & Annappa, 2017*; *Mertens, 2020*; *Haisjackl & Weber, 2010*). For example, *Ghattas, Soffer & Peleg (2014)* aim to avoid customers rejecting delivery in a bottle manufacturing process. In the domain of healthcare, examples of undesired outcomes are

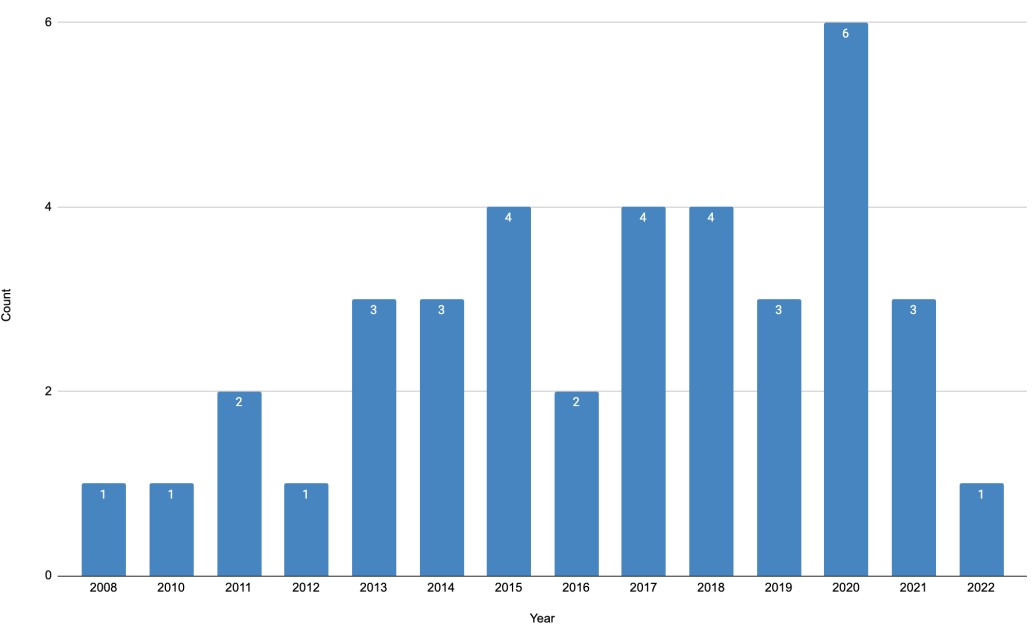

**Figure 1** **Distribution of papers per publication year.**

patients entering a critical stage (*Thomas, Kumar & Annappa, 2017*), or medical mistakes due to patient restrictions (*Mertens, 2020*).

The second main objective considers optimizing the process efficiency. Most papers consider optimizing temporal perspectives (15 out of 21), such as cycle or processing time. More specifically, in *Wibisono et al. (2015)*; *Kim, Obregon & Jung (2013)*; *Obregon, Kim & Jung (2013)*; *Thomas, Kumar & Annappa, (2017)*, reducing cycle time is defined as the main objective. In this context, reducing cycle time means that the cycle time should be gradually improved by ensuring that each coming case takes less time than the previous average. Thus, it differs from the previously described objective of avoiding deadline violation. For instance, *Thomas, Kumar & Annappa, (2017)* describe a method to minimize the cycle time of an environmental permit application process. Others focus on processing time, *i.e.,* time spent by a resource resolving a task (*Dumas et al., 2018*). For instance, in *Park & Song (2019)* the aim is to reduce the processing time of manual tasks in a loan application process. Another set of methods aims at reducing the defect rate. For example, a method aims to reduce the likelihood and severity of a fault based on risk prediction (*Conforti et al., 2015*). Two papers (*Goossens, Demewez & Hassani, 2018*; *Terragni & Hassani, 2019*) describe methods that aim to improve revenues, *e.g.*, by increasing customer lifetime value (*Goossens, Demewez & Hassani, 2018*). Finally, two methods (*Khan et al., 2021*; *Schonenberg et al., 2008*) can be utilized for different targets. As such, a method by *Schonenberg et al. (2008)* provides an example of optimizing the cycle time but the authors suggest that it could also be used to optimize costs, quality, or utilization.

## Prescribed interventions

Prescriptive process monitoring methods prescribe actions to take, *i.e.,* interventions. These interventions can be categorized according to the process perspective of the prescribed intervention. Our review indicates that interventions commonly concern control flow and resource perspectives.

A common intervention perspective is control flow, such as prescribing the next task to perform (*de Leoni, Dees & Reulink, 2020*; *Heber, Hagen & Schmollinger, 2015*; *Nakatumba, Westergaard & van der Aalst, 2012*). More specifically, in *de Leoni, Dees & Reulink (2020)*, the next best task is prescribed to the professional who helps a customer find a new job, whereas in *Weinzierl et al. (2020b)*, the next step is presented to the end-user. Following the prescribed intervention can improve execution time, customer satisfaction, or service quality. In other studies, a sequence of next steps is prescribed as an intervention (*Detro et al., 2020*; *Yang et al., 2017*; *Triki et al., 2013*). For instance, in one method, the appropriate treatment of a blood infusion is prescribed for patients based on their personal information (*Detro et al., 2020*), whereas another method prescribes steps to be taken in a trauma resuscitation process (*Yang et al., 2017*). Such interventions aim to improve treatment quality.

Another group of methods focuses on the resource perspective, *e.g.*, which resource should perform the next task. For instance, *Wibisono et al. (2015)* prescribe which police officer is best suited for the next task in a driving license application process based on their predicted performance. In another method, a mechanic is recommended to carry out car repairs because s/he is predicted to finish within a defined time given her/his schedule and expertise (*Sindhgatta, Ghose & Dam, 2016*).

Some papers propose prescribing multiple interventions for one case (*Shoush & Dumas, 2021*; *Nezhad & Bartolini, 2011*; *Barba, Weber & Valle, 2011*). For example, an intervention to make an offer to a client is prescribed together with a suggestion for a specific clerk to carry out the task (*Shoush & Dumas, 2021*). Similarly, in an IT service management process, recommending the next task and the specialist to perform it can help to resolve open cases faster (*Nezhad & Bartolini, 2011*).

When reviewing the identified papers, we noted that interventions can be divided into two categories: intervention frequency and intervention purpose. Intervention frequency describes when interventions are prescribed. In this regard, prescriptive monitoring methods can be continuous or discrete. If a method is continuous, it prescribes an intervention for multiple or all activities of an ongoing case. For example, prescribing the best-suited resource for each next task (*Wibisono et al., 2015*). Discrete interventions, in comparison, prescribe actions to be taken only when a need is detected. For instance, in *Metzger, Kley & Palm (2020)*, interventions are triggered only when it is detected that the probability of a negative outcome exceeds a defined threshold.

The intervention purpose describes whether a method is optimizing or guiding. On the one hand, optimizing methods suggest interventions to improve an ongoing case with respect to a certain performance measure. Such interventions are sometimes based on predictions (*e.g.*, *Gröger, Schwarz & Mitschang (2014)*; *de Leoni, Dees & Reulink (2020)*; *Fahrenkrog-Petersen et al. (2022)*). Thus, a method predicts the outcomes of an ongoing

case and then prescribes an intervention. Optimizing methods can be correlation- or causality-based. For instance, in *Khan et al. (2021)*, the proposed method prescribes interventions based on predictions from past execution data that is labeled with outcomes. To this end, the method uses past execution data to train a model that correlates possible interventions with the likelihood of their effectiveness. Similarly, in a method by *Gröger, Schwarz & Mitschang (2014)*, decision trees are used to recommend an intervention, where a categorized metric value is correlated with selected attributes of a process instance, and a methods by *Ghattas, Soffer & Peleg (2014)* provides predictions of process instance performance and evaluates the predictions by a domain expert.

Conversely, in a method by *Shoush & Dumas (2021)*, the probability of an undesired outcome is estimated, as well as the impact of a given intervention on the outcome of a case. Thus, the method first estimates the probability of the undesired outcome. Then, a causal model is built with the purpose to estimate the conditional average treatment effect (CATE) of an intervention in a given case. In the context of process outcome optimization, the CATE is the increase (or decrease) in the probability of a positive (or negative) case outcome. In the context of process efficiency optimization, the CATE is the increase (or decrease) in the efficiency of the process. Similarly, in a method by *Bozorgi et al. (2021)*, a causal estimation module is utilized to estimate the effect of a given intervention on the running case. In *Teinemaa et al. (2018)*; *Fahrenkrog-Petersen et al. (2022)*, the concept of mitigation effectiveness of an intervention is introduced which helps to estimate the relative benefit of an intervention at a certain point in time.

On the other hand, guiding methods provide recommendations solely based on an analysis of historical traces. In guiding methods, a set of actions are prescribed based on the similarity rate of a current ongoing case and previous cases (*Terragni & Hassani, 2019*; *Triki et al., 2013*; *Sindhgatta, Ghose & Dam, 2016*). For instance, in a method by *Terragni & Hassani (2019)* steps in a customer journey are proposed based on calculating the similarity of the current journey to the journeys that lead to a desirable KPI value in the past.

### Required data input & feature encoding

Our review shows that prescriptive process monitoring methods use control flow, resource, temporal, and domain-specific data. Some methods focus on a single type of data, while other methods combine data input of different types.

As expected, methods that prescribe interventions impacting control flow, such as the next task to execute, commonly use control flow data. For example, in *Conforti et al. (2015)*, the authors apply decision trees on data, such as task duration and frequency, to predict the risk of a case fault, *e.g.*, exceeding the maximum cycle time and cost overrun. In a similar manner, *Goossens, Demewez & Hassani (2018)* prescribe the next task by using the sequence of events as a key feature.

Data on resources are used to trigger interventions related to different prescription perspectives. For instance, one method utilizes the execution time of past resource performance to reallocate pending work items to resources with higher efficiency (*Yaghoibi & Zahedi, 2017*). In another method, the authors use resource roles and capabilities combined with domain-specific features, such as vehicle type, to recommend which

mechanic should be assigned the next task (*Sindhgatta, Ghose & Dam, 2016*). The data on resources is thus used to predict which resource would improve the probability of the vehicle repair being finished within a defined time.

Temporal data, *e.g.*, day of the week, is also used to prescribe interventions. Such data is commonly used in combination with other data, such as control flow or resource data. For instance, the best-suited resource to execute the next task is recommended utilizing the period of the day (morning, afternoon, or evening), inter-arrival rate, and task queue data as input (*Wibisono et al., 2015*). In another method (*Bozorgi et al., 2021*), temporal information (month, weekday, hour) of the last event and the inactive period before the most recent event in the log are used to evaluate the effectiveness of an intervention to reduce cycle time.

Domain-specific data, such as materials used in a manufacturing process (*Ghattas, Soffer & Peleg, 2014*), patient demographics, and treatment attributes in a patient treatment process (*Yang et al., 2017*), are also utilized as data input. For instance, data on previously treated patients and data on a current patient are used to assess the predicted outcome of alternative next tasks (*Mertens, 2020*). This method recommends the task that has the best-predicted outcome for a patient to reduce the risk of medical mistakes, such as prescribing the wrong medication.

## Modeling technique

Prescriptive process monitoring methods utilize a range of modeling techniques. These techniques are typically different for the two different types of intervention purpose: optimizing or guiding (as discussed in the section on Prescribed Interventions).

Several optimizing methods make use of techniques such as decision trees (*Sindhgatta, Ghose & Dam, 2016*; *Kim, Obregon & Jung, 2013*; *Obregon, Kim & Jung, 2013*; *Conforti et al., 2015*; *Ghattas, Soffer & Peleg, 2014*). For example, *Sindhgatta, Ghose & Dam (2016)* use decision tree learning to predict the performance of an ongoing case considering process context and historical trace. Then, the k-nearest neighbor (kNN) approach is used to determine which process context values are likely to lead to the desired outcomes. Similarly, *Kim, Obregon & Jung (2013)* use decision trees to predict indicators such as remaining flow time or total labor cost. On this basis, their algorithm recommends the best resources to minimize completion time or total labor cost. Another technique used for prediction is neural networks (*Metzger, Kley & Palm, 2020*; *Weinzierl et al., 2020b*; *Park & Song, 2019*). For instance, *Metzger, Kley & Palm (2020)* use recurrent neural networks (RNNs) with Long Short-Term Memory (LSTM) cells to compute predictions and reliability estimates. The algorithm predicts deviations that indicate the need for potential intervention. Similarly, *Weinzierl et al. (2020b)* compare three configurations of LSTMs (without context, with context, and with embedded context) to suggest the next clicks to a customer in a sales order process. Other examples of modeling techniques that are used for prediction include support vector machines (*Goossens, Demewez & Hassani, 2018*). The authors utilize this technique to predict the next event to occur in a customer journey. Based on the prediction, actions that optimize a chosen KPI, are recommended to the customer.

The majority of guiding methods in our review use nearest neighbor to provide recommendations (*Mertens, 2020*; *Haisjackl & Weber, 2010*; *Arias et al., 2018*; *Arias, Munoz-Gama & Sepúlveda, 2016*; *Nezhad & Bartolini, 2011*; *Triki et al., 2013*; *Yang et al., 2017*). For example, *Haisjackl & Weber (2010)* propose a method that compares a current partial trace with past traces in an event log and recommends interventions with regard to the chosen performance goal. Similarly, *Arias et al. (2018)* introduce a similarity-based method for resource allocation. More specifically, the authors examine the expertise and workload of a resource based on past executions. As a result, by allocating resources to perform the next tasks in a loan application process, the processing time (*i.e.,* time spent by a resource resolving a task) is reduced. Some similarity-based methods do not utilize any specific modeling technique. Rather, they implement an independent method that provides recommendations based on specific criteria. For instance, the method proposed by *Abdulhameed et al. (2018)* computes co-working relationships of resources based on frequency (how often a resource performed a task in the past) and duration (what the processing time was). As a result, the algorithm recommends resources that have the best working harmony with other resources.

We also note that there are papers that explain in detail the approach they take and the algorithms they develop for their method (76% of papers). Such detail includes, for example, describing all components of the predictor module as well as the generation of prescriptions (*e.g.*, *Shoush & Dumas (2021)*; *Fahrenkrog-Petersen et al. (2022)*). There are, however, papers that explain the approach in abstract terms and give references to techniques they use but do not describe them in detail (24% papers) (*e.g.*, *Barba, Weber & Valle, 2011*; *Obregon, Kim & Jung, 2013*).

## Policy

In prescriptive process monitoring methods, the policy describes the conditions required to prescribe an intervention. Our review shows that similarity-based methods use—as expected—a similarity-based policy, while prediction-based methods employ a range of policies (*e.g.*, set of rules, probability of a negative outcome above a threshold).

With the similarity-based policy, an intervention is prescribed based on the similarity of the current case to previous cases (*e.g.*, *Thomas, Kumar & Annappa, 2017*; *Triki et al., 2013*; *Nezhad & Bartolini, 2011*; *Yang et al., 2017*). For example, *Triki et al. (2013)* utilize a similarity-based policy for recommending steps to take in a disaster management process. The algorithm is implemented in a disaster management system and is designed to compare the current case with previous cases. In an example of an IT service management system, the authors (*Nezhad & Bartolini, 2011*) compare the running case with resolved cases to recommend the best next steps to handle the running case.

One policy that prediction-based methods employ is the probability of a negative outcome exceeding a defined threshold (*Fahrenkrog-Petersen et al., 2022*; *Teinemaa et al., 2018*; *Metzger, Kley & Palm, 2020*; *Shoush & Dumas, 2021*). More specifically, when a certain value is predicted to be higher than a set threshold, it signals that an intervention is required. However, these methods evaluate the probability in combination with other conditions. As such, the method proposed by *Shoush & Dumas (2021)* first predicts the

 

[3]Link to the full framework: https://doi.org/10.6084/m9.figshare.19455806.v3.

probability of an undesired outcome in a loan application process. Then, it checks the resource availability and cost of carrying out the intervention. As a result, an intervention (*e.g.*, making more offers to clients) and a clerk to execute the task are prescribed in the loan application process to the cases that would most benefit from it. In *Fahrenkrog-Petersen et al. (2022)*, the proposed method evaluates the probability of a negative outcome together with a cost model (*i.e.,* computing costs of an intervention, undesired outcome, and compensation) and the mitigation effectiveness.

Several methods are adapted to multiple types of policies, *i.e.,* the policy can change depending on the purpose (*Huber, Fietta & Hof, 2015*; *Bozorgi et al., 2021*; *Heber, Hagen & Schmollinger, 2015*). For example, in *Huber, Fietta & Hof (2015)*, the policy can be time-, deadline-, decision-, goal-based, or combined. Thus, if their method is used to reach a specific goal (*e.g.*, increase customer satisfaction), the goal-based policy is used to propose suitable interventions. Similarly, the deadline-based policy is used to prescribe interventions when a deadline violation is detected. In methods described by *Bozorgi et al. (2021)* and *Heber, Hagen & Schmollinger (2015)*, the policy is user-defined, *i.e.,* the user can set up criteria based on the purpose they pursue. For instance, the process worker defines the percentage of customers who should receive a phone call (*Bozorgi et al., 2021*).

Other examples of policy are exceeding the defined metric limits (*Gröger, Schwarz & Mitschang, 2014*; *Weinzierl et al., 2020a*), *i.e.,* prescribing interventions when a deviation of a metric limit is detected. For instance, the method by *Gröger, Schwarz & Mitschang (2014)* monitors the values of a target metric and prescribes changing the resource settings when a deviation from that target is identified. Several methods also define maximum metric improvement as policy (*de Leoni, Dees & Reulink, 2020*; *Nakatumba, Westergaard & van der Aalst, 2012*; *Goossens, Demewez & Hassani, 2018*). For instance, in *Nakatumba, Westergaard & van der Aalst (2012)*, the next best action is recommended based on the highest predicted value in relation to a goal. Others compose a set of rules which serve as a policy (*Arias et al., 2018*; *Abdulhameed et al., 2018*; *Barba, Weber & Valle, 2011*). As such, *Abdulhameed et al. (2018)* define a rule for recommending a resource based on the resource's availability, processing time, and compatibility with other resources.

## FRAMEWORK

Having reviewed the papers, we followed the research questions to identify the characteristics of the prescriptive process monitoring methods we identified. Starting from **RQ₁**, we clustered the methods according to their objective. We then followed the remaining research questions to detail the clustering. As a result, the proposed framework (Figs. 2 and 3) characterizes prescriptive process monitoring methods based on ten characteristics. The framework[3] reads from left to right and starts with the objective of using a prescriptive process monitoring method. It then goes on to describe the interventions to reach that objective, the input the method requires, the techniques it uses, and the policy to trigger interventions.

| Objective | Target | Prescription perspective | Intervention | Input perspective | Feature encoding | Modeling technique | Policy | Intervention frequency | Intervention purpose | Detailed algorithm | Reference(s) |
|---|---|---|---|---|---|---|---|---|---|---|---|
| Optimizing process outcome | Temporal outcome | Resource | Resource settings | All available | All case attributes | Decision trees | Exceeding metric limits | Continuous | Optimizing (correlation) | Y | Gröger et al. (2014) |
| | | | Resource allocation | R; T; D | Resource experience, preference, collaboration, utilization; domain-specific case attributes; time of day | Decision trees (prediction), nearest neighbor (to select values) | Highest predicted metric value | Continuous | Guiding | Y | Sindhgatta et al. (2016) |
| | | Control flow | Next task to perform | T; C | Timestamps; activity sequence | Composite classification model | Multiple options (time-based, deadline-based, decision-based, goal-based) | Continuous | Optimizing (correlation) | N | Huber et al. (2015) |
| | | | | C | Activity sequence, activity duration | LSTM | Exceeding metric limits, maximum metric improvement | Discrete | Optimizing (correlation) | Y | Weinzierl et al. (2020a) |
| | | | | Metric specific | Metric specific features | Random Forest, SVM, decision trees | Maximum metric improvement | Continuous | Optimizing (correlation) | Y | de Leoni et al. (2020) |
| | Categorical outcome | Varies (multiple) | An alarm to trigger an intervention | All available | All case attributes | Random forest, gradient boosted trees, empirical thresholding | Probability of a negative outcome above a threshold (for k consecutive events), cost model (intervention, undesired outcome, compensation costs), mitigation effectiveness | Discrete | Optimizing (causality) | Y | Fahrenkrog-Petersen et al. (2022); Teinemaa et al. (2018) |
| | | | Various interventions | All available | All case attributes | Ensemble deep supervised learning (RNN, LSTM), online reinforcement learning | Probability of a negative outcome above a threshold, cost model (adaptation, penalty, compensation costs), reliability estimate | Discrete | Optimizing (causality) | Y | Metzger et al. (2020) |
| | | | Various interventions and resources | All available | All case attributes | XGBoost (prediction), ORF (estimation) | Probability of a negative outcome above a threshold, intervention cost, resource availability | Discrete | Optimizing (causality) | Y | Shoush et al. (2021) |
| | | | Various process decisions | D; C | Task sequence, domain-specific case attributes | Decision trees | Highest predicted metric value | Continuous | Optimizing (correlation) | Y | Ghattas et al. (2014) |
| | | | Next task to perform and the resource | D; C; R | Task sequence, past resource performance, domain-specific case attributes | ModCNN, longest common subsequence | Similarity based | Discrete | Guiding | Y | Thomas et al. (2017a) |
| | | Control flow | Next task to perform | D | Domain-specific case features | Nearest neighbor | Similarity based | Continuous | Guiding | N | Mertens et al. (2020) |
| | | | Next task to perform | C | Past process executions, enabled activities | Nearest neighbor | Similarity based | Discrete | Guiding | Y | Haisjackl et al. (2010) |

**Figure 2** Prescriptive process monitoring framework: optimize process outcome objective.

## Framework components

The main characteristic of the framework is the Objective. Our analysis shows that the identified methods can be divided into two categories according to the objective they pursue (section 'Objective'). The first category aims to reduce the percentage of cases with a negative outcome, *i.e.,* optimize the process outcome (Fig. 2). Methods in the second category aim to optimize process efficiency which is captured *via* a quantitative performance metric defined at the level of each case, *e.g.*, cycle time (Fig. 3). The next characteristic of the framework is the Target: the metric used to assess if the performance is improved by a prescribed intervention. For the objective of optimizing the process outcome, the target may be a count of a categorical case outcome (*e.g.*, customer complaints) or of a temporal outcome (deadline violations). Conversely, quantitative performance targets include quantifiable measures such as cycle time, labor cost, or revenue.

The next two characteristics (Prescription Perspective, Intervention) capture the interventions that a method prescribes to pursue a defined objective. Thus, the Prescription Perspective describes to which process aspect an intervention is related, *e.g.*, resource or control flow. We also included the category "multiple" for methods that describe several interventions. Then, the characteristic Intervention lists the actual prescribed interventions (see section 'Interventions'). For instance, actual intervention can be which resource to assign to the next task.

The following four characteristics define the data (Input Perspective, Feature Encoding), techniques (Modeling Technique), and policies (Policy) to trigger interventions. Namely, the Input Perspective describes the types of features, *i.e.,* input data, required for a specific

| Objective | Target | Prescription perspective | Intervention | Input perspective | Feature encoding | Modeling technique | Policy | Intervention frequency | Intervention purpose | Detailed algorithm | Reference(s) |
|---|---|---|---|---|---|---|---|---|---|---|---|
| Optimizing process efficiency | Cycle time | Resource | Resource for the next task | C; T | Queue, inter-arrival rate, timeframe | Naive Bayes model (prediction), Naive Bayes selection rule (selection) | Highest predicted resource performance | Continuous | Optimizing (correlation) | Y | Wibisono et al. (2015) |
| | | | | R; T | Co-working history, past resource performance | Model-less | Set of rules (resource availability, resource compatibility, processing time) | Continuous | Guiding | Y | Abdulhameed et al. (2018) |
| | Processing time | | Resource allocation | R; C; T | Activity sequence, processing time, start time, finish time, instance weight | LSTM | Scheduling algorithm | Continuous | Optimizing (correlation) | Y | Park et al. (2019) |
| | | | | R | Resource experience, past resource performance, resource workload | Nearest neighbor | Set of rules (resource availability, resource experience, processing time, processing cost) | Continuous | Guiding | Y | Arias et al. (2018) |
| | | | Resource allocation | R | Resource experience, past resource performance, resource workload, resource capabilities | Nearest neighbor | Set of rules (individual and in-team performance) | Continuous | Guiding | Y | Arias et al. (2016) |
| | Labor cost | | Resource for the next task | R; C | Activity sequence, processing time, labor cost | Decision trees | Highest predicted resource performance | Continuous | Opimizing (correlation) / Optimizing (correlation) | Y / N | Kim et al. (2013); Obregon et al. (2013) |
| | Cycle time | Control flow | Task reassignment | R; C; T | Activity sequence, arrival rate, resource responsibilities, resource workload, past | Statistical analysis | Scheduling algorithm | Continuous | Guiding | Y | Yaghoubi et al. (2017) |
| | | | Next task to perform | C | Activity sequence, past process performance | Model-less | Maximum metric improvement | Continuous | Guiding | N | Nakatumba et al. (2012) |
| | | | | C; T; D | Activity sequence occurence, activity duration, amounts of damage | State model | User-defined function | Continuous | Guiding | N | Heber et al. (2015) |
| | Revenue | | | C | Activity sequence | SVM | Maximum metric improvement + similarity | Continuous | Optimizing (correlation) | Y | Goossens et al. (2018) |
| | | | | C | Domain-specific case features, activity sequence | kNN, ALS, BPR | Similarity based | Continuous | Guiding | Y | Terragni et al. (2019) |
| | Defect rate | | Set of tasks to perform | D; C | Domain-specific case features | Decision trees | Similarity based + internal and external rules | Continuous | Guiding | Y | Detro et al. (2020) |
| | | | | D; C | Domain-specific case attributes | Nearest neighbor | Similarity based | Continuous | Guiding | Y | Yang et al. (2017) |
| | | | | D; C | Domain-specific case attributes | Nearest neighbor | Similarity based | Continuous | Guiding | N | Triki et al. (2013) |
| | | | Next task to perform | C; R | Task durations and frequencies, resources | Decision trees (prediction), mixed-integer linear programming (distribution) | Probability and severity of risk, scheduling algorithm | Discrete | Optimizing (correlation) | Y | Conforti et al. (2015) |
| | | | Next clicks in the process | D; C | Activity sequence, domain-specific case features | LSTMs | Probability above a threshold | Continuous | Guiding | Y | Weinzierl et al. (2020b) |
| | Varies (multiple) | | Best next path | D; C | Activity sequence, domain-specific case features | DCw-MANN | Maximum metric improvement | Discrete | Optimizing (correlation) | Y | Khan et al. (2021) |
| | | | Next task to perform | C; T | Activity sequence, processing time | Constraint programming | Set of rules (constraints) | Discrete | Optimizing (correlation) | Y | Schonenberg et al. (2008) |
| | Cycle time | Varies (multiple) | Next task to perform and resource | R; C; D | Activity sequence, resources, domain-specific case features (case type, priority, status, tags) | Nearest neighbor | Similarity based | Continuous | Guiding | N | Nezhad et al. (2011) |
| | | | | C; R | Activity frequency, processing time | Decision trees | Similarity based | Continuous | Guiding | N | Thomas et al. (2017b) |
| | | | | C; R | Activity duration, resource availability, number of instances | Constraint programming | Set of rules (optimized plans, resource availability) | Continuous | Guiding | N | Barba et al. (2011) |
| | | | Various interventions | T; R | Temporal information of the last event, workload, difference between start time of the case and start time of the timeframe | Orthogonal random forests | User-defined policy | Discrete | Optimizing (causality) | Y | Bozorgi et al. (2021) |

**Figure 3** **Prescriptive process monitoring framework: optimize process efficiency objective.**

method (see section 'Required Data Input & Feature Encoding'). Thus, the categories we elicited are (C)ontrol flow (*e.g.*, activities, sequence, and frequencies), (R)esources (*e.g.*, performers of activities), (T)emporal features (time-related), and (D)omain-specific (features that depend on the domain or type of process). The characteristic Feature Encoding explains how features are further refined by a prescriptive method. For instance, resource-perspective features can be encoded as resource experience, resource performance, or resource workload.

The characteristic Modeling Technique relates to the technique used to predict the outcome of a process or its performance based on the input (section 'Modeling Technique'). Next, Policy relates to the conditions under which an intervention is prescribed (see section 'Policy'). For example, under a similarity-based policy, an intervention is prescribed based on the similarity of the current case to previous cases. Some policies come in the form of a set of rules. For example, a policy can consist of two rules. First, the need for an intervention is detected when the probability of a negative outcome exceeds a defined threshold. Second, the effectiveness of the intervention is assessed before prescribing it.

The characteristic Intervention Frequency shows whether a method is continuous (prescribes actions at every step) or discrete (only when needed). Additionally, Intervention Purpose describes whether a method is optimizing or guiding and whether the optimizing method is based on correlation or causality (see section 'Interventions').

Another characteristic in the framework is the categorization of methods with regard to the detailed description of the utilized algorithms (Detailed Description of Algorithm) (see section 'Modeling Technique'). In this boolean category, we mark papers that provide sufficient detail on the method or an algorithm with an ''Y''. We perceive details to be sufficient when a step-by-step explanation is provided that allows reproducing the described method or algorithm. Respectively, papers that only give references to approaches they use but do not provide any detail are marked with an ''N''.

Finally, the characteristic Example (see the full version of the framework) can be used as a reference to how the introduced method with its inputs, technique, and policies was used to trigger interventions to reach the objective of a process in a specific domain.

### Framework usage

As an example, the framework could be utilized to explore existing methods from the objective, target, and prescribed process perspective. As such, if one *e.g.*, seeks to minimize the number of temporal outcome violations (*e.g.*, reduce the number of cases that violate a deadline), the aim is to optimize the process outcome (Objective), more specifically, temporal outcome (Target). The framework shows that this can be achieved by prescribing interventions related to the control flow or resources of the respective process (Prescription Perspective). If we follow the control flow perspective, the framework shows that a set of methods can recommend, for example, next task to perform in a running case (Intervention) (*Huber, Fietta & Hof, 2015*; *Weinzierl et al., 2020a*; *de Leoni, Dees & Reulink, 2020*). However, these methods have different input perspectives and they utilize different modeling techniques. Continuing with selecting control flow from the input perspective, leads to an optimizing method that relies on causality (Intervention Purpose) and prescribes discrete interventions (Intervention Frequency). This method uses LSTM to predict the next actions and then conducts an evaluation of them and prescribes the one that is calculated to have the optimal value for the future course of a process instance (*Weinzierl et al., 2020a*).

## RESEARCH GAPS AND IMPLICATIONS

The presented framework provides an overview of existing prescriptive process monitoring methods by categorizing them according to their objectives and targets. The framework also presents different available methods and the different ways these methods enable reaching particular objectives. The overview, however, also unveils several gaps and associated implications for future research.

First, we observe that the majority of studies tested their methods with synthetic and/or real-life event logs. Therefore, the validation is done using a real-world or synthetic observational event log, but crucially it is not done in real-life settings. An attempt to test the effectiveness of interventions in real-life settings was made by *Dees et al. (2019)*. Their

study showed that predictions were rather accurate, but the interventions did not lead to desired outcomes. Thus, the proposed methods should be validated in real-life settings to ensure their usefulness in practice (*Márquez-Chamorro, Resinas & Ruiz-Cortéz, 2018*).

Second, our review showed that the majority of methods focus on identifying cases in which interventions should be applied and finding the point in time an intervention should be triggered during the execution of a case. In contrast, little attention has been paid to the problem of discovering which interventions could be applied to optimize a process with respect to a performance objective. Discrete methods leave it up to the users (stakeholders) to define possible intervention(s) a priori (*e.g.*, *de Leoni, Dees & Reulink, 2020*; *Teinemaa et al., 2018*). Methods that use observational event logs from BPIC, rely on winner reports to identify possible interventions (*e.g.*, *Shoush & Dumas, 2021*; *Teinemaa et al., 2018*). Continuous methods, in contrast, focus on recommending the next task(s) (*e.g.*, *Goossens, Demewez & Hassani, 2018*; *Yang et al., 2017*) or resource allocation (*e.g.*, *Arias, Munoz-Gama & Sepúlveda, 2016*; *Abdulhameed et al., 2018*; *Kim, Obregon & Jung, 2013*). Thus, one direction for further research could be to design methods that support the discovery of interventions from business process event logs, textual documentation, or other unstructured or structured process metadata.

Related to the above problem of discovering interventions, we observed that methods differ with regard to the nature of prescriptions. As our review shows, there is a body of methods that focus on providing prescriptions to guide a user during process execution (*e.g.*, *Terragni & Hassani, 2019*; *Weinzierl et al., 2020b*). These methods are commonly based on the similarity of an ongoing case with previous executions of the same process. At the same time, there are methods that specifically optimize processes according to a performance measure (*e.g.*, *Khan et al., 2021*; *Metzger, Kley & Palm, 2020*). Thus, methods differ between simply guiding the user and providing a specific suggestion for an action to be taken next. On this basis, future researchers can make a distinction for the type of method and, respectively, prescriptions they are developing. As such, the spectrum of options starts with guiding methods which require developing an algorithm to compare the ongoing case with the past executions which ended in desired outcomes to recommend the suitable option for continuing the running case. Another option is a category of methods that are optimizing. Such methods would typically require incorporating predictions based on past executions. One way is to consider the correlation of a predicted value with the likelihood of it having a positive effect on the running case. Another way is to focus on causality, where the impact of the proposed intervention on the outcome of the case should be estimated—for example, through building a causal estimation model (*Shoush & Dumas, 2021*; *Bozorgi et al., 2021*).

Another gap in existing research relates to the problem of designing and tuning policies for prescriptive process monitoring. Existing prediction-based methods (*e.g.*, *Sindhgatta, Ghose & Dam, 2016*; *Gröger, Schwarz & Mitschang, 2014*) prescribe an intervention when the probability of a negative outcome exceeds a defined threshold. However, because predictive models are based on correlation (as opposed to causal relations), the prescriptions produced might not address the cause of a negative outcome or poor performance (*e.g.*, the cause of delay). In this respect, we note that only a few existing methods take causality

into account when designing policies (*e.g.*, *Shoush & Dumas, 2021*; *Bozorgi et al., 2021*). Thus, developing policy design techniques that take causality into account can be another direction for further research.

As discussed in *Dees et al. (2019)* and *de Leoni, Dees & Reulink (2020)*, the choice of whether or not to apply an intervention or the choice of which intervention to apply often depends on contextual factors. Some interventions may prove ineffective or counter-productive, for example, due to second-order effects (*i.e.,* an intervention has a consequence that has a subsequent consequence). For example, an intervention wherein a customer is contacted pro-actively in order to prevent a complaint may actually increase the probability of a complaint (*Dees et al., 2019*). Similarly, assigning a resource to a case that is running late might lead to other cases being neglected, thus creating delays elsewhere and subsequently resulting in a higher ratio of delayed cases. Detecting such second-order effects requires human judgment and iterative policy validation (*e.g.*, *via* A/B testing (*Kohavi & Longbotham, 2017*)). In this respect, it is striking that existing prescriptive process monitoring methods do not take into account the need to interact with human decision-makers. A crucial step in this direction is the ability to explain why a prescriptive monitoring system recommends a given intervention. There are two aspects to this question. First, explaining the prediction that underpins a given prescription (prediction explanation), and second, explaining the policy that is used to trigger a prescription (policy explanation). A possible direction to enhance the applicability of prescriptive monitoring methods in practice is to integrate explainability mechanisms. While several proposals have been made to enhance the explainability of predictive process monitoring methods (*Hsieh, Moreira & Ouyang, 2021*; *Rizzi et al., 2022*), the question of policy explanation in the area of prescriptive process monitoring is—to the best of our knowledge—unexplored. In other words, current methods do not incorporate mechanisms to explain why an intervention is recommended for a given case and in a given state.

Besides the aforementioned gaps, our findings also highlight that the majority of methods in the field of prescriptive process monitoring aim to improve processes along the temporal perspective (*e.g.*, cycle time, processing time, deadline violations (*Abdulhameed et al., 2018*; *Park & Song, 2019*)). In comparison, other performance dimensions are only represented in a few cases (defect rate in *Detro et al., 2020*; *Conforti et al., 2015*; *Yang et al., 2017*, revenue in *Goossens, Demewez & Hassani, 2018*; *Terragni & Hassani, 2019*). Thus, another research direction could be to investigate other performance objectives that could be enhanced *via* prescriptive process monitoring.

Finally, our review also highlights a lack of common terminology in the field. This might be due to the novelty of the research field of prescriptive process monitoring. Authors use a large variety of labels. As such, the terms "proactive process adaptation" (*Metzger, Kley & Palm, 2020*), "on-the-fly resource allocation" (*Wibisono et al., 2015*), "next-step recommendation" (*Huber, Fietta & Hof, 2015*; *Nezhad & Bartolini, 2011*) are all used to describe the development and application of prescriptive process monitoring methods. This highlights the need for common terminology.

In summary, we note the following research gaps:

- Validating methods in real-life settings to ensure their usefulness in practice.
- Designing methods to support the discovery of interventions from event logs and assessing their potential effectiveness.
- Developing policy design techniques that take causality into account.
- Improving explainability of prescriptive process monitoring methods.
- Investigating other performance objectives rather than temporal that could be enhanced *via* prescriptive process monitoring.
- Developing a common terminology in the field.

### Threats to validity

The SLR methodology is associated with certain limitations and threads to validity (*Kitchenham & Charters, 2007*; *Ampatzoglou et al., 2019*). First, there is a potential risk of missing relevant publications during the search. We mitigated this risk by conducting a two-phase search that included a broad range of key terms, as well as backward referencing. Another potential threat is to exclude relevant publications during screening. We mitigated this threat by using explicitly defined inclusion and exclusion criteria. Furthermore, all unclear cases were examined and discussed by all authors of this paper. Third, there is a threat of data extraction bias as this step involves a degree of subjectivity. We discussed each paper in the final list and refined the data extraction when needed to minimize this risk.

## CONCLUSION

Our article provides an overview of the research on prescriptive process monitoring and outlines a framework for categorizing methods in this field. The framework categorizes existing methods according to their objective, target metric, intervention type, technique, data input, and policy used to trigger interventions.

Our findings indicate that existing prescriptive process monitoring methods primarily have one of two objectives: optimizing process outcome or optimizing process efficiency. In order to achieve the respective objective, a range of interventions can be prescribed. Our review indicates that the interventions commonly concern control flow and resource perspectives. We also note that interventions can be divided into two categories: intervention frequency (when interventions are prescribed) and intervention purpose (how interventions are prescribed). According to the intervention frequency characteristic, the corresponding methods can be continuous (prescribing an intervention for multiple or all activities of an ongoing case) or discrete (an intervention is triggered only when a need is detected). As to intervention purpose, the interventions can be optimizing or guiding. On the one hand, optimizing methods suggest interventions to improve an ongoing case with respect to a certain KPI or performance measure. Such interventions can be correlation- or causality-based. On the other hand, guiding methods provide recommendations solely based on an analysis of historical traces.

Our review also shows that existing prescriptive process monitoring methods use control flow, resource, temporal, and domain-specific data as required input. This data is processed by modeling techniques which, as we found, differ based on whether a method is optimizing

or guiding. Similarly, the policies to prescribe an intervention differ based on the method. As such, guiding methods use similarity-based policies, while for optimizing methods the policies can differ (*e.g.*, set of rules, maximum metric improvement, probability of a negative outcome above a threshold, etc.).

We also identified research gaps and associated research avenues to highlight where the field of prescriptive process monitoring is headed (*i.e.,* answering the question *"Quo vadis?"*). In particular, our work highlighted: (i) a lack of *in vivo* validation of the proposed methods; (ii) a lack of methods for discovering suitable interventions and assessing their potential effectiveness; (iii) little emphasis on explainability and feedback loops between a prescriptive monitoring system and its end-users; and (iv) a narrow focus on temporal metrics and comparatively little work on applying prescriptive monitoring to other performance dimensions. While our results might not be surprising for process mining experts, they provide an overview of the current state of the art in the field of prescriptive process monitoring and give insights for future research directions.

### Funding
This research is funded by the Estonian Research Council (PRG1226) and the European Research Council (PIX Project). The funders had no role in study design, data collection and analysis, decision to publish, or preparation of the manuscript.

### Grant Disclosures
The following grant information was disclosed by the authors:
The Estonian Research Council: PRG1226.
The European Research Council (PIX Project).

### Competing Interests
The authors declare there are no competing interests.

### Author Contributions
- Kateryna Kubrak conceived and designed the experiments, performed the experiments, analyzed the data, prepared figures and/or tables, authored or reviewed drafts of the article, and approved the final draft.
- Fredrik Milani conceived and designed the experiments, analyzed the data, authored or reviewed drafts of the article, and approved the final draft.
- Alexander Nolte conceived and designed the experiments, analyzed the data, authored or reviewed drafts of the article, and approved the final draft.
- Marlon Dumas conceived and designed the experiments, authored or reviewed drafts of the article, and approved the final draft.

### Data Availability
The raw data is available in the article and at Figshare:

- section Method, page 5: Kubrak, Kateryna (2022): Review Protocol_Prescriptive Process Monitoring: Quo Vadis?. figshare. Dataset. https://doi.org/10.6084/m9.figshare.19455572.v2

- section Framework, Figures 2 and 3: Kubrak, Kateryna (2022): Prescriptive process monitoring framework.xlsx. figshare. Journal contribution. https://doi.org/10.6084/m9.figshare.19455806.v3.

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
