# Peer review of "Prescriptive process monitoring: Quo vadis?"

_PeerJ Computer Science, doi:10.7717/peerj-cs.1097_

## Round 0.1 · original submission · Major Revisions

Dear authors,

I attach reviews of the above paper that you submitted to PeerJ Computer Science for publication. As you can read, some issues have been raised by the reviewers and so your article must be revised before publication can be considered. Your revision will be sent out for a second review. Please carefully address the issues raised in the comments. I'm sure that with a major revision, the criticisms raised by the reviewers can be met.

In particular, please take into account the comments concerning the level of maturity of the chosen field for performing not only a technically correct literature review, but also an insightful one. I do share some of these concerns and therefore invite you to (1) consider the idea of maintaining the methodological instrument of a literature review to focus on the production of a framework as suggested by a reviewer and (2) carefully justify (or assess) the limitations of some of the studies included in the survey, I would kindly invite you also to assess the minor (but annoying) issue of the labels of bibliographic references.

When sending in the next revision, please be sure to provide a letter, explicitly stating how all these criticisms have been dealt with.

Reviewer 1 ·

Basic reporting

Dear Editor

The proposed study aims at providing a systematic literature review of prescriptive process monitoring which is a relatively new topic in the field of process mining. Through the systematic approach proposed in [Kitchenham, B. A. and Charters, S. (2007)], the study identify a set of 36 relevant paper which are later analyzed and classified with respect to a multi-dimensional framework of the prescriptive process monitor method proposed in each paper reviewed. The features analyzed in the framework are: the objective of the method, the process intervention prescribed by the method, the type of data required, the technique used, and the policy used to prescribe interventions.
I recommend the paper for publication in JPeer since it appears to me as the only review, in this newly developed field, considering all types of potential interventions in process-aware methods. As such it serves as a valid introduction to the subject and gives a detailed overall picture of the state of the art. Moreover, the systematic analysis provides insight into the field, indicates some gaps in the current development and some potential areas for future research, providing some value also to the field experts.
Overall, the article is well-written and clear: the purpose of the review, the review protocol, the research questions and the classification criteria are clearly described.
In order to improve the readability further I suggest the following minor edits which mainly deal with the display of some results:
1. In table 1 I find some ambiguities in the definition of Exclusion Criteria. It seems to me that the descriptions of these criteria are not homogeneous: the answer required for the paper to be excluded is either YES or NO inconsistently. As an example, “Is the paper language English?” requires NO as an answer for the paper to be excluded, contrarily the question “Is the paper a duplicate?” requires a YES as an answer for the paper to be excluded. Even though the ambiguity is solved in the text I would suggest the authors to modify the descriptions in the table to increase readability of the paper. A possible way is to rephrase the description as statements rather than questions. In the Inclusion Criteria this problem is not present since all three questions requires the answer YES for the paper to be included.
2. In fig. 2 and 3 the last column is dedicated to the paper reference number (or paper ID). However, this number, differently from the arxiv version, is no more displayed in the current style of the bibliography. This makes the results difficult to read without the use of the online resources. Since these figures summarize the main result of the review, I would suggest the authors to make explicit the relation between this column and the bibliography, this should increase the readability of the results, and the handiness of the paper.
3. Add citation for A/B testing (row 487)

Experimental design

no comment

Validity of the findings

no comment

Reviewer 2 ·

Basic reporting

The article "Prescriptive Process Monitoring: Quo Vadis?" attempts to conduct a systematic literature review to examine the current applications for prescriptive process analytics. After a detailed presentation of the research methodology for the literature review, the authors propose a framework to summarise the identified studies in terms of different aspects such as performance objectives, modeling approaches, types of interventions and strategies, etc. The review is within the scope of the journal.

The language/formal presentation of the manuscript is at a decent level.

One of the main problems is that a technical error affects the readability of the derived results. In the last columns of Figures 2 and 3, which summarise the main contribution of the submitted paper, the authors have given numbers to the identified papers. However, in the citation style of the submitted manuscript, the bibliography is sorted by the last name of the first author. It is therefore impossible to deduce from these numbers which papers the authors are talking about. This format-specific problem, which is easy to fix, has nevertheless complicated the review process, as it has been necessary to find the identified studies between the lines. However, this is not the only main concern of the paper.

Experimental design

The problem studied is very relevant and appealing, especially given the growing interest of the academic and industrial communities in adopting machine learning-based approaches for predictive process monitoring. The paper takes this further by examining the studies that not only predict various targets of interest but also seek to define the course of action to prevent undesirable outcomes. The literature search, the coding/filtering of the literature, and the descriptive analysis of the identified papers are solid and well introduced. Despite the well-elaborated methodological part, the main contribution of the paper falls behind expectations. The reasons for this are manifold. (see 1. Basic Reporting and 3. Validity of findings)

Validity of the findings

A quick analysis of the identified papers shows that while some thematically pertinent and interesting studies such as:

Fahrenkrog-Petersen, S. A., Tax, N., Teinemaa, I., Dumas, M., de Leoni, M., Maggi, F. M., and Weidlich,M. (2022). Fire now, fire later: alarm-based systems for prescriptive process monitoring.Knowl. Inf. Syst., 64(2):559–587

de Leoni, M., Dees, M., and Reulink, L. (2020). Design and evaluation of a process-aware recommender system based on prescriptive analytics. In2nd International Conference on Process Mining, ICPM, pages 9–16. IEEE

were identified, the relevance and maturity of some studies are not necessarily convincing.

For instance, a thorough inspection of the following article:

Huber, S., Fietta, M., and Hof, S. (2015). Next step recommendation and prediction based on process mining in adaptive case management. InS-BPM ONE, pages 3:1-3:9. ACM .

reveals that its authors provide almost no details on the algorithmic background of the adopted predictive process monitoring approach, the mechanism of intervention generation and evaluation, the details of the underlying process mining use case and data, and so on. Only a mockup prototype of the dashboard is presented, which is not necessarily useful for the target group of the literature review conducted.

Another identified study,

Triki, S., Saoud, N. B. B., Dugdale, J., and Hanachi, C. (2013). Coupling case-based reasoning and process mining for a web-based decision support system for crisis management. InWETICE, pages 245-252. IEEE Computer Society

is just an in-progress research article investigating the combination of conventional case-based reasoning approaches and process mining. A predictive analytics scenario, as defined at the beginning of the submitted review paper, is not found here. The intervention scenarios are also missing.

Or, the following article:

Weinzierl, S., Stierle, M., Zilker, S., and Matzner, M. (2020b). A next click recommender system for web-based service analytics with context-aware lstms. In HICSS, pages 1-10. ScholarSpace.

is a good contribution to predictive process analytics research, however, the "prescriptive" analytics component is lacking in this study. This only uses LSTM to make predictions about the next steps. It is not necessarily understood why the authors of the submitted review paper classify it as a prescriptive process analytics study.

This list of not necessarily relevant articles labeled as prescriptive process analytics studies can be easily extended. This is due to the fact that the definitions of predictive and prescriptive process analysis are sometimes vague. The authors have already mentioned the term "intervention" several times throughout the paper, which constitutes the crucial difference in this regard. However, an intuition or experience-based definition of an intervention by experts or even scholars does not necessarily make the article or application prescriptive. Various algorithmic approaches from different fields such as mathematical programming and optimization, simulation, evolutionary computation, explainable artificial intelligence, causal learning, inference, etc. should be superimposed on predictive process analysis techniques to produce the intervention mechanisms and policies efficiently, robustly, and precisely.

There is also a need for quantitative and qualitative assessment concepts for the soundness of such interventions. A prominent example is as follows:

Bertsimas, D., & Kallus, N. (2020). From predictive to prescriptive analytics. Management Science, 66(3), 1025-1044.

Unfortunately, very few of the identified studies meet the quality requirements to be classified as prescriptive process analytics. As mentioned earlier, the submitted paper is thematically relevant but not necessarily timely relevant, as there is still a need for good quality studies to lay the groundwork for a review paper. On these grounds, I would encourage the authors to consider modifying the structure of their article from a literature review of prescriptive process analytics to building a conceptual and mathematical framework for developing and evaluating interventions for process analytics by transferring experience from other fields.

Additional comments

Finally, there are some details that I think should be revised. First, in the subsection "Prescribed goals of process monitoring," the author lists the first category as "reducing the error rate." This designation can be misleading, especially considering that there are some process analytics approaches from manufacturing. This label is confusing and may be too generic considering the elements it includes. Another problem in the same subsection concerns the definition of the objectives. The authors should explain what the main difference is between "avoid exceeding the allowed limits for cycle times" and "reducing cycle time". These objectives are mapped into two different categories. The only difference I see here is the nature of the prediction result in terms of ML modeling, with the first category describing a (binary) classification problem and the second category applying to regression problems. However, there is not much difference in the objectives and underlying interventions. The taxonomy lacks precision.

In summary, the article introduces an interesting topic, provides a solid methodology, and delivers a good descriptive overview. However, I think that the findings and the practical/scientific implications need to be significantly improved. Therefore, I vote for a major revision.

---

## Round 0.2 · Minor Revisions

The reviewer(s) have recommended publication, but also suggest some minor revisions to your manuscript. Please carefully consider these comments when revising your paper.

Please describe the changes and responses to the reviewer comments, in detail, in a separate file. You should also create a version of your revised manuscript that highlights the changes within the document by using the track changes mode in MS Word.

Reviewer 1 ·

Basic reporting

In this reviewed version of the paper the authors have solved the issues I had previously highlighted. Mainly table 1 and fig 2 and fig 3 are now more readable than before.

Moreover, following the indications of the second reviewer the authors have improved the framework used to classify the reviewed articles adding two dimensions of analysis: the first one is the intervention purpose which clusters the methods as optimizing or guiding, and whether an optimizing method is based on correlation or causality. The second dimension added to the framework is a Boolean value indicating if the reviewed article contains the details about the algorithm it presents. In my opinion this improves the robustness of the review, making easier for the reader to identify the significance of each paper.
The authors also inserted some new paragraph to better describe the features used in the framework.
Finally, the authors added to the review a new paper that has been published in the meanwhile, during the review phase, and that satisfy the SLR criteria.

In conclusion I think that the paper has been improved by these modifications and, following the conclusions of my first review, I recommend it for publication in PeerJ.

I list here some misprints I encountered during the reading
-row 174: insude
-row 327: utilizies
-fig 3, column Target, row 7: vaires
-fig 3, column Policy, row 16: thershold
-row 678, 740: Francescomarino, C. D. should be Di Francescomarino, D. as in row 720

Experimental design

no comment

Validity of the findings

no comment

Reviewer 2 ·

Basic reporting

In their revised manuscript, the authors have attempted to resolve the issues highlighted in the first round of reviews. The clarifications and adjustments addressed in the relevant sections appear to be reasonable. Eliminating potential sources of misunderstanding and sharpening terminology have improved the quality of the manuscript. In addition, references have been formatted appropriately.

Although the quality and relevance of some identified studies are questionable, the need for a systematic review of prescriptive process analytics, in general, is well elaborated. Several relevant studies have been added to the revised version.

Experimental design

I have a few more remarks:

- In their response document, the authors discussed the vague definition of prescriptive analytics and provided some different examples. I strongly recommend that the authors provide a discussion either in the introduction or in the background section on how they define prescriptive analytics, the main elements of prescriptiveness, and its components and limitations. Such an introduction is missing.

- The results illustrated in Figures 2 and 3 are presented as image files or figures. However, these should be converted into tables. If the main results are presented in an inappropriate format, their value is diminished, especially by limiting the search process.

- The definition of "causality" is also important. Indeed, as the authors also discussed, the main elements of causal explanations are the intervention and the outcomes after the defined treatments. Some identified studies categorized as "causality-based optimization" are correct in this sense. However, for some studies, e.g., Kim et al. (2013), Obregon et al. (2013), and Park et al. (2019), it is not clear how these studies examined causality. In my opinion, they do not do so at all. The authors should provide a separate discussion for each publication.

Validity of the findings

In my subjective opinion, publishing a review article on prescriptive process analysis is still premature. On the other hand, this article may be considered acceptable if such an article is needed now. Overall, I switch my decision to acceptance subject to minor changes listed in the bullet points above.

---

## Author Rebuttal · Round 0.2

**Summary of Changes**

**Title "Prescriptive Process Monitoring: Quo Vadis?"
(manuscript id: #CS-2022:03:71927)**

Dear editor and reviewers,

We would like to thank you for providing us with constructive reviews and for encouraging us to submit a revised manuscript. Our interpretation of the reviews is that the paper covers a topic that has a growing interest in the community and is methodologically sound. The main concerns discussed by the reviewers to our understanding relate to the timeliness of the systematic literature review, as well as limitations of some studies included in the review, and the definitions of terms covered by the review. Below we provide a detailed response to the comments we received on our original submission and the changes that were made to address these comments. In short, we have made the following changes:

- Elaborated on the differences between the identified methods to explain the range of interventions that they prescribe through adding the categorization based on intervention frequency and intervention purpose (subsection Prescribed Interventions)
- Labeled the methods according to the level of detail they provide in the description of their algorithmic approach (subsection Modeling Technique)
- Clarified the naming of the categories of objectives (subsection Prescriptive Process Monitoring Objectives)
- Included the newly emerged insights from the points above into suggestions for further work (section Research Gaps and Implications)
- Addressed the issue of the referencing of the studies in the framework (framework file on figshare, figures 2 and 3)

We believe that these changes led to an improved paper, and we hope these improvements are evident in the response and the paper itself. We would like to express our gratitude to the entire review team and are looking forward to your feedback.

Yours sincerely,

- the authors

## Reviewer 1

| Reviewer's comments | Author's response |
|---|---|
| I recommend the paper for publication in JPeer since it appears to me as the only review, in this newly developed field, considering all types of potential interventions in process-aware methods. As such it serves as a valid introduction to the subject and gives a detailed overall picture of the state of the art. Moreover, the systematic analysis provides insight into the field, indicates some gaps in the current development and some potential areas for future research, providing some value also to the field experts. <br><br> Overall, the article is well-written and clear: the purpose of the review, the review protocol, the research questions and the classification criteria are clearly described. | Thank you. Please find our detailed responses below. |
| In table 1 I find some ambiguities in the definition of Exclusion Criteria. It seems to me that the descriptions of these criteria are not homogeneous: the answer required for the paper to be excluded is either YES or NO inconsistently. As an example, "Is the paper language English?" requires NO as an answer for the paper to be excluded, contrarily the question "Is the paper a duplicate?" requires a YES as an answer for the paper to be excluded. Even though the ambiguity is solved in the text I would suggest the authors to modify the descriptions in the table to increase readability of the paper. A possible way is to rephrase the description as statements rather than questions. In the Inclusion Criteria this problem is not present since all three questions requires the answer YES for the paper to be included. | Thank you for this comment and for pointing this out. We have made the change so it is now consistent. Specifically, we changed the formulation of the exclusion and inclusion criteria in the table in Section "Method" (page 4), as well as in the full review protocol attached as supplementary material. With these changes, if the answer to an exclusion criterion is "no", then the paper is excluded (e.g., EC1 The paper is digitally accessible. -> no -> exclude). Likewise, for an inclusion criterion, if the answer is "yes", the paper is included (e.g., IC1 The paper is relevant to the domain of prescriptive process monitoring. -> yes -> include) |
| In fig. 2 and 3 the last column is dedicated to the paper reference number (or paper ID). However, this number, differently from the arxiv version, is no more displayed in the current style of the bibliography. This makes the results difficult to read without the use of the online resources. Since these figures summarize the main result of the review, I would suggest the authors to make explicit the relation between this column and the bibliography, this should increase the readability of the results, and the handiness of the paper. | Indeed, thank you. We now see that such mapping of included papers can complicate the review process. We have adapted the framework file and figures 2 and 3 (pages 11 and 12) to match the reference list in the paper. |

| | |
|---|---|
| Add citation for A/B testing (row 487) | Fixed. Citation to the relevant source added (page 13, line 536). |

## Reviewer 2

| Reviewer's comments | Author's response |
|---|---|
| The problem studied is very relevant and appealing, especially given the growing interest of the academic and industrial communities in adopting machine learning-based approaches for predictive process monitoring. The paper takes this further by examining the studies that not only predict various targets of interest but also seek to define the course of action to prevent undesirable outcomes. The literature search, the coding/filtering of the literature, and the descriptive analysis of the identified papers are solid and well introduced. Despite the well-elaborated methodological part, the main contribution of the paper falls behind expectations. The reasons for this are manifold. (see 1. Basic Reporting and 3. Validity of findings) | Thank you for your insightful comments which helped us to improve the paper. Please find our detailed responses below. |
| A quick analysis of the identified papers shows that while some thematically pertinent and interesting studies such as: <br><br> Fahrenkrog-Petersen, S. A., Tax, N., Teinemaa, I., Dumas, M., de Leoni, M., Maggi, F. M., and Weidlich,M. (2022). Fire now, fire later: alarm-based systems for prescriptive process monitoring.Knowl. Inf. Syst., 64(2):559–587 <br><br> de Leoni, M., Dees, M., and Reulink, L. (2020). Design and evaluation of a process-aware recommender system based on prescriptive analytics. In2nd International Conference on Process Mining, ICPM, pages 9–16. IEEE <br><br> were identified, the relevance and maturity of some studies are not necessarily convincing. <br> For instance, a thorough inspection of the following article: <br><br> Huber, S., Fietta, M., and Hof, S. (2015). Next step recommendation and prediction based on process mining in adaptive case management. InS-BPM ONE, pages 3:1-3:9. ACM . <br> reveals that its authors provide almost no details on the algorithmic background of the adopted predictive process monitoring approach, the mechanism of intervention generation and evaluation, the details of the underlying process mining use case and data, and so on. Only a mockup prototype of the dashboard is presented, which is not necessarily useful for the target group of the literature review conducted. | Thank you for this comment. We understand this concern. The availability of a detailed algorithm is not among the inclusion/exclusion criteria. However, we do acknowledge that not having the opportunity to study the algorithm step-by-step is a clear limitation in a paper that proposes a prescriptive monitoring method with a computational component. To capture this important limitation of some of the reviewed studies, we introduced a new category to our framework, namely "Detailed Description of Procedure". In this category, we mark with "Y'' (yes) those papers that provide sufficient detail on the method or an algorithm to reproduce it. Conversely, papers that give references to approaches they use but do not provide a step-by-step explanation are marked with "N'' (no). We have also provided explanations about this categorization on pages 9 and 11). |

Another identified study,

Triki, S., Saoud, N. B. B., Dugdale, J., and Hanachi, C. (2013). Coupling case-based reasoning and process mining for a web-based decision support system for crisis management. InWETICE, pages 245-252. IEEE Computer Society

is just an in-progress research article investigating the combination of conventional case-based reasoning approaches and process mining. A predictive analytics scenario, as defined at the beginning of the submitted review paper, is not found here. The intervention scenarios are also missing.

Or, the following article:

Weinzierl, S., Stierle, M., Zilker, S., and Matzner, M. (2020b). A next click recommender system for web-based service analytics with context-aware lstms. In HICSS, pages 1-10. ScholarSpace.

is a good contribution to predictive process analytics research, however, the "prescriptive" analytics component is lacking in this study. This only uses LSTM to make predictions about the next steps. It is not necessarily understood why the authors of the submitted review paper classify it as a prescriptive process analytics study.

This list of not necessarily relevant articles labeled as prescriptive process analytics studies can be easily extended. This is due to the fact that the definitions of predictive and prescriptive process analysis are sometimes vague. The authors have already mentioned the term "intervention" several times throughout the paper, which constitutes the crucial difference in this regard. However, an intuition or experience-based definition of an intervention by experts or even scholars does not necessarily make the article or application prescriptive. Various algorithmic approaches from different fields such as mathematical programming and optimization, simulation, evolutionary computation, explainable artificial intelligence, causal learning, inference, etc. should be superimposed on predictive process analysis techniques to produce the intervention mechanisms and policies efficiently, robustly, and precisely.

Thank you for this very valuable point. Indeed, you are right in saying that the definitions of predictive and prescriptive process analysis are sometimes vague. As we have also seen from our review, there is also a lack of uniformity regarding the terms used in the literature in this field.

We analyzed the papers again with the goal of eliciting what the different authors mean by "prescriptive". Following this analysis, we noted that two distinct notions of "prescriptiveness" can be found across the reviewed studies, depending on the purpose of the prescriptions. On the one hand, some of the reviewed approaches provide execution guidance (e.g. what could be the next activity in a case) by analyzing historical cases that are similar to an ongoing case. The purpose of these approaches is to indicate to caseworkers, what would be the most "usual" course of action in a given state of a case. These methods for not aim at optimizing the performance of the process with respect to some performance metrics or desired outcomes. In contrast, the second category of approaches explicitly aims at optimizing process performance. This latter category of approaches can be further divided into two sub-categories. The first sub-category of approaches prescribes actions on the basis of predictive models, specifically models that predict the most likely outcome of a case. These approaches do not make a distinction between correlation and causation -- the prescriptions they make are based on correlations between actions and desired outcomes or performance measures (e.g. cases where a given action is performed finish on-time more often than cases where this action

| | is not performed). The second sub-category of approaches makes a distinction between correlation and causation and they generate prescriptions based on causal relations between an intervention and a desired outcome or performance metric. |
| | Accordingly, we added a column corresponding to the "intervention purpose". This column categorizes the papers into either "guiding prescriptions", "optimizing prescriptions [correlation], or "optimizing prescriptions [causality]". Furthermore, we have added a new paragraph in the Discussion section to highlight this new insight (page 13, line 505). |

There is also a need for quantitative and qualitative assessment concepts for the soundness of such interventions. A prominent example is as follows:

Bertsimas, D., & Kallus, N. (2020). From predictive to prescriptive analytics. Management Science, 66(3), 1025-1044.

Unfortunately, very few of the identified studies meet the quality requirements to be classified as prescriptive process analytics. As mentioned earlier, the submitted paper is thematically relevant but not necessarily timely relevant, as there is still a need for good quality studies to lay the groundwork for a review paper. On these grounds, I would encourage the authors to consider modifying the structure of their article from a literature review of prescriptive process analytics to building a conceptual and mathematical framework for developing and evaluating interventions for process analytics by transferring experience from other fields.

Thank you for this suggestion. Looking closer into this comment led us to identify that there are three distinct notions of prescriptiveness in the literature in this field. We understand your concern that this review of prescriptive process monitoring methods is coming during an early stage in the development of the field. On the other hand, we believe it is important to have a snapshot of the literature at this stage so that the authors can refer to it and build on top of state-of-the-art instead of reinventing the wheel. Also, a review at this point in the development of this field can help the research community to uniformize the terminology and to have a shared understanding of recurrent concepts. This is an active field, as evidenced by the fact that in the past few months, since we completed the first version of this manuscript, one new relevant paper was published [1] (we added it to our SLR for this revision), and two others were posted in arXiv [2, 3]. Your proposal is interesting but, at this time, it is unfortunately beyond the scope of this paper.

| | |
|---|---|
| Finally, there are some details that I think should be revised. First, in the subsection "Prescribed goals of process monitoring," the author lists the first category as "reducing the error rate." This designation can be misleading, especially considering that there are some process analytics approaches from manufacturing. This label is confusing and may be too generic considering the elements it includes. | We agree that this naming is misleading. We have addressed this issue by changing the names of both objectives. We now distinguish between the objective of optimizing the process outcome and the objective of optimizing the process efficiency. The objective of optimizing the process outcome relates to ensuring that the process outcome is positive. This objective is commonly expressed with binary metrics. The second objective of optimizing the process efficiency relates to a particular quantitative aspect of process performance (e.g., reducing the cycle time). We have updated the framework and the text in the section Prescriptive Process Monitoring Objectives accordingly (page 6, line 216). |
| Another problem in the same subsection concerns the definition of the objectives. The authors should explain what the main difference is between "avoid exceeding the allowed limits for cycle times" and "reducing cycle time". These objectives are mapped into two different categories. The only difference I see here is the nature of the prediction result in terms of ML modeling, with the first category describing a (binary) classification problem and the second category applying to regression problems. However, there is not much difference in the objectives and underlying interventions. The taxonomy lacks precision. | Thank you for pointing out this unclarity. We have elaborated on the difference between the two in subsection "Prescriptive Process Monitoring Objectives" (page 6, line 226). More specifically, we have explained that the first target (temporal outcome) refers to reducing deadline violations (y/n), whereas the second target is concerned specifically with reducing the cycle time (the case takes less time on average). |
| In the last columns of Figures 2 and 3, which summarise the main contribution of the submitted paper, the authors have given numbers to the identified papers. However, in the citation style of the submitted manuscript, the bibliography is sorted by the last name of the first author. It is therefore impossible to deduce from these numbers which papers the authors are talking about. This format-specific problem, which is easy to fix, has nevertheless complicated the review process, as it has been necessary to find the identified studies between the lines. | Thank you for pointing this out. We understand that such mapping of included papers might have complicated the review process and apologize for that. We have adapted the framework file to match the references list in the paper. |

# References

[1] Khan, A., Le, H., Do, K., Tran, T., Ghose, A., Dam, H. K., and Sindhgatta, R. (2021). Deepprocess: Supporting business process execution using a mann-based recommender system. In ICSOC, volume 13121 of Lecture Notes in Computer Science, pages 19–33. Springer.

[2] Branchi, S., Di Francescomarino, C., Ghidini, C., Massimo, D., Ricci, F., & Ronzani, M. (2022). Learning to act: a Reinforcement Learning approach to recommend the best next activities. arXiv preprint arXiv:2203.15398.

[3] Agarwal, P., Gupta, A., Sindhgatta, R., & Dechu, S. (2022). Goal-Oriented Next Best Activity Recommendation using Reinforcement Learning. arXiv preprint arXiv:2205.03219.

---

## Round 0.3 · accepted · Accept

Dear authors, thanks for addressing all the reviewers comments,

Concerning the appropriateness of publishing a review on prescriptive BPM now, I have to say that I fully understand - and somehow share - the comment of one of the reviewers, on the not fully mature nature of prescriptive BPM.

Nonetheless I truly believe that the careful review on prescriptive BPM presented in the paper is a valuable contribution to this field and will help its consolidation, which perhaps may lead to more mature SRLs in the future.

Congratulations!